# Calibration and Validation of a Transfer Radiometer Applied to a Radiometric Benchmark Transfer Chain

**Kaichao Lei** [1,2], **Xin Ye** [1,*], **Nan Xu** [3], **Shuqi Li** [1], **Yachao Zhang** [1], **Yuwei Wang** [1], **Zhiwei Liu** [3] **and Zhigang Li** [1,*]

[1] Changchun Institute of Optics, Fine Mechanics and Physics, Chinese Academy of Sciences, Changchun 130033, China
[2] School of Optoelectronics, University of Chinese Academy of Sciences, Beijing 100049, China
[3] National Institute of Metrology, Beijing 100029, China
[*] Correspondence: yexin@ciomp.ac.cn (X.Y.); lizhg@ciomp.ac.cn (Z.L.)

**Abstract:** A transfer radiometer (TR) applied to an on-orbit radiometric benchmark transfer chain has been developed, which can achieve the high-precision transformation of power and radiance responsivity and transmit the radiance responsivity traced to the cryogenic radiometer to remote sensors, such as an imaging spectrometer, so that the on-orbit remote sensors can achieve the high accuracy calibration of $10^{-3}$ magnitude. Radiance comparison experiments between the TR and the radiance standard of the National Institute of Metrology (NIM) were carried out to demonstrate the absolute accuracy of the TR radiance measurement. At 780.0 nm and 851.9 nm, the relative measurement uncertainties of the TR filter-free channel were 0.24% (k = 1). Additionally, the radiance measurement results of the TR were consistent with those of the NIM radiance meter, and the radiance measurement results' relative differences between the TR and the NIM radiance meter were approximately 0.04% at 780.0 nm and 851.9 nm. The relative measurement uncertainties of TR 780.4 nm and 851.8 nm filter channels were 0.89% (k = 1) and 0.84% (k = 1), respectively. Additionally, the radiance measurement results of the TR 780.4 nm and 851.8 nm filter channels were consistent with the radiances of the integrating sphere source calibrated by the NIM at 780.4 nm and 851.8 nm; the relative differences between the radiances measured by the two TR filter channels and the radiances of the integrating sphere source itself were better than 0.56%. This proved that the TR could measure the monochromatic source radiance with a measurement uncertainty of 0.24% and measure the broadband source radiance with a measurement uncertainty better than 0.89%. The TR can be applied to the radiometric benchmark transfer chain to improve the measurement precision of on-orbit remote-sensing instruments.

**Keywords:** transfer radiometer; traceable; radiance comparison; cryogenic radiometer

## 1. Introduction

The Earth's climate is changing silently, and clarifying the causes of climate change and predicting future climate change are considerable challenges facing the scientific community today. Space remote-sensing technology is the main means for people to obtain information about the Earth's climate; however, the measurement accuracy of contemporary remote-sensing instruments is insufficient to help people obtain accurate climate information. Therefore, there is an increasingly urgent demand for the high-precision radiometric calibration of space on-orbit remote-sensing instruments. The radiometric calibration process of remote sensors includes pre-launch laboratory calibration and post-launch on-orbit calibration. Even though the laboratory calibration methods can accomplish high accuracy calibration, remote sensors are affected by harsh conditions such as high pressure and vibration during the launch, as well as the impact of solar radiation and cosmic particles for extended periods during the on-orbit operation, which degrades the performance of remote sensors, ultimately reducing the measurement accuracy. Therefore, the on-orbit

calibration of remote sensors is necessary. At present, the main methods of on-orbit radiation calibration rely on solar diffusers and standard lamps. However, the measurement accuracy of contemporary on-orbit calibration methods is low, and it is challenging to meet the application needs of quantitative remote sensing in environmental monitoring, weather forecasting, and other fields. Therefore, it is highly desirable to establish an on-orbit radiation reference with higher accuracy and traceability to the International System of Units (SI). To improve the measurement accuracy of on-orbit remote sensors, the United States and Europe have proposed spacecraft missions, including CLARREO and TRUTHS, respectively, which can provide on-orbit SI-traceable calibration [1–6]. The Chinese Spaced-Based Radiometric Benchmark (CSRB) project, established with the goal of launching a radiometric benchmark satellite adopting a new on-orbit calibration system, has been under development since 2014. The Space Cryogenic Absolute Radiometer (SCAR) is used as the onboard radiation benchmark. The Earth–Moon Imaging Spectrometer (EMIS) is used to measure the reflected solar spectral radiance. The spectral radiance responsivities of the EMIS and other remote-sensing instruments is calibrated by SCAR via the benchmark transfer chain (BTC) and cross-calibration [7,8]. The TR is the core instrument of the BTC. Firstly, the TR spectral power responsivity is directly traced to the SCAR by measuring the same laser beam with the SCAR. Additionally, the power responsivity is transformed into the radiance responsivity according to its transformation coefficient of power responsivity and radiance responsivity. Then, by measuring the same quasi-Lambert source, the TR transfers the radiance responsivity traceable to the SCAR to the EMIS and other remote sensors to achieve high-accuracy radiance calibration of EMIS and other remote sensors.

Currently, transfer radiometers are mainly used to calibrate the radiance of the sources used for optical sensor calibrations and automated vicarious calibrations when they have been calibrated by SIRCUS or other calibration devices [9–19]. The TR proposed in this paper is mainly used for the on-orbit radiometric benchmark transfer chain, which can measure both power and radiance and achieve a high-precision transformation of power responsivity and radiance responsivity. The TR can transfer the radiance responsivity traceable to the SCAR to the EMIS and other remote sensors. The design of the TR and a monochromatic source radiance comparison experiment at 852.1 nm have been described previously [20]. Based on previous work, this paper further verifies the accuracy of the TR monochromatic and broadband radiance measurement by experiments. This paper mainly introduces the radiance measurement comparison experiments between the TR and the radiance standard of the NIM. We performed narrowband, detector-based power responsivity calibration and radiance measurement comparison experiments of the filter-free channel of the TR and a broadband, source-based radiance measurement comparison experiment of the TR. In this paper, we first briefly describe the TR structure and function. Then, we describe the experimental method, experimental equipment, uncertainty analysis, and comparison results of the narrowband, detector-based, power responsivity calibration, and radiance measurement comparison experiment of the TR filter-free channel and broadband, source-based, radiance measurement comparison experiment of TR filter channels.

## 2. Introduction to the Transfer Radiometer

The TR functions as a bridge connecting the SCAR and EMIS and transfers the radiance responsivity traceable to the SCAR to the EMIS, realizing high-accuracy radiance calibration of the EMIS. The design of the TR has been described previously [20]. The TR is mainly composed of a radiance-measuring tube, a filter wheel, an integrating sphere, a Si photodiode, and an extended InGaAs detector. The radiance-measuring tube determines the field of view of the TR and completes the transformation of power responsivity and radiance responsivity. The radiance-measuring tube can also suppress stray light, and the TR stray light suppression ability is better than 0.06%. The TR has 11 filter channels, and a filter-free channel is obtained by rotating the filter wheel. The power responsivity of the TR filter-free channel is calibrated by a cryogenic radiometer. Additionally, the TR filter-free channel transforms power responsivity to radiance responsivity. The spectral radiance

responsivity of the TR filter channel is derived through the filter transmittance curve measured in the radiance measurement mode and the spectral radiance responsivity of the filter-free channel. The TR filter-free channel is mainly used to measure monochromatic sources such as lasers, and the TR filter channel is primarily used to measure broadband sources such as halogen tungsten lamps. Radiance measured by the TR is obtained according to the radiance responsivity of the TR and its output value. The integrating sphere is coated with PTFE and has three exit ports. The radiance-measuring tube, Si photodiode, and extended InGaAs detector are installed at the three exit ports. A Si photodiode combined with an InGaAs detector can receive optical signals in the spectral wavelength range of 380 nm~2350 nm. Considering the large dark background signal of the InGaAs detector, an SR860 lock-in amplifier combined with an SR540 optical chopper and an amplifier are used to measure the output signal of the InGaAs detector. The original idea was to use a picoammeter (Keithley 6485) to measure the output signal of the Si photodiode. However, it was difficult to determine the SR540 optical chopper blade position after the SR540 optical chopper was turned off, and the SR540 optical chopper blade easily blocked the transmitted radiation into the optical port of the TR. Therefore, the output signal of the Si photodiode was also measured by the SR860 lock-in amplifier combination with the SR540 optical chopper and an amplifier to facilitate the measurement.

## 3. Narrowband, Detector-Based, Radiometric Calibration, and Comparison Experiment

Transferring the radiance responsivity traced to the SCAR to the EMIS is the main role of the TR in BTC. This experiment was conducted to validate the precision of the TR filter-free channel radiance responsivity.

### 3.1. Narrowband, Detector-Based, Power Responsivity Calibration

The experimental setup is shown in Figure 1. Compared with prior power responsivity calibration and radiance comparison experiments [20], some experimental components were updated. Firstly, this experiment replaced the integrating sphere. The diameter of the integrating sphere was 360 mm, and the exit port diameter was 100 mm. There was a rotatable baffle with a diameter of 30 mm inside the integrating sphere. The laser was directly incident on the baffle. The rotary baffle was used to suppress the laser speckle effects. Secondly, the radiance meter of the NIM was updated. The first precision aperture of the NIM radiance meter was not located at the exit port of the integrating sphere, and the distance between the first precision aperture and the exit port of the integrating sphere was 1048.9 mm. The diameter of the first precision aperture was 30.074 mm. The second precision aperture was placed next to the trap detector. The diameter of the second precision aperture was 4.022 mm. The distance between the first and second precision aperture was 722.7 mm. Finally, the new measurement system was composed of an SR860 lock-in amplifier, an SR540 optical chopper, and an amplifier used to measure the output signal of the TR, instead of a picoammeter.

In the experimental optical path, plane mirror 1 and plane mirror 2 were used to turn the optical path. Plane mirror 2 is a reversible plane mirror. In the optical path of the power responsivity calibration experiment, plane mirror 2 was removed from the optical path, and the light was directly received by the trap detector through plane mirror 1 and the beam shrinking device. The beam shrinking device consisted of two lenses and a spatial filter placed between the two lenses. The main function of the beam shrinking device was to ensure that the TR and trap detector received the same beam. The trap detector calibrated by a cryogenic radiometer was used as the power standard in the TR power responsivity calibration experiment [21,22]. The experiment light source was a continuous-wave tunable laser. The laser gain medium of the tunable laser was Ti:Sapphire, and the tunable laser line width was less than 4 MHz. The tunable laser mode structure was a single longitudinal mode. The tunable laser wavelength was adjusted to 780.0 nm and 851.9 nm based on wavelength meter feedback. Similar to a previous study [20], it was necessary to guarantee that the TR responsivity calibration experiment and the radiance comparison experiment

light source wavelength were the same. The laser beam emitted from the tunable laser passed through the power stabilizer, plane mirror 1, and the beam shrinking device and was received by the trap detector. A voltmeter measured the trap detector output signal. When the voltmeter output voltage and the trap detector power responsivity were known, the laser power was derived. After the measurement of the trap detector was completed, the trap detector was removed to ensure that it did not block the laser beam. The TR optical axis coincided with the laser beam by moving the TR. The TR measured the laser beam using the filter-free channel. The measurement system consisting of the SR860 lock-in amplifier, SR540 optical chopper, and amplifier measured the output signal of the TR. The TR filter-free channel power responsivity was derived from the measured output voltage as a function of measured laser power. The radiance responsivity of the TR at 780.0 nm and 851.9 nm was obtained according to its transformation coefficient of power responsivity and radiance responsivity. Table 1 shows the experimental results.

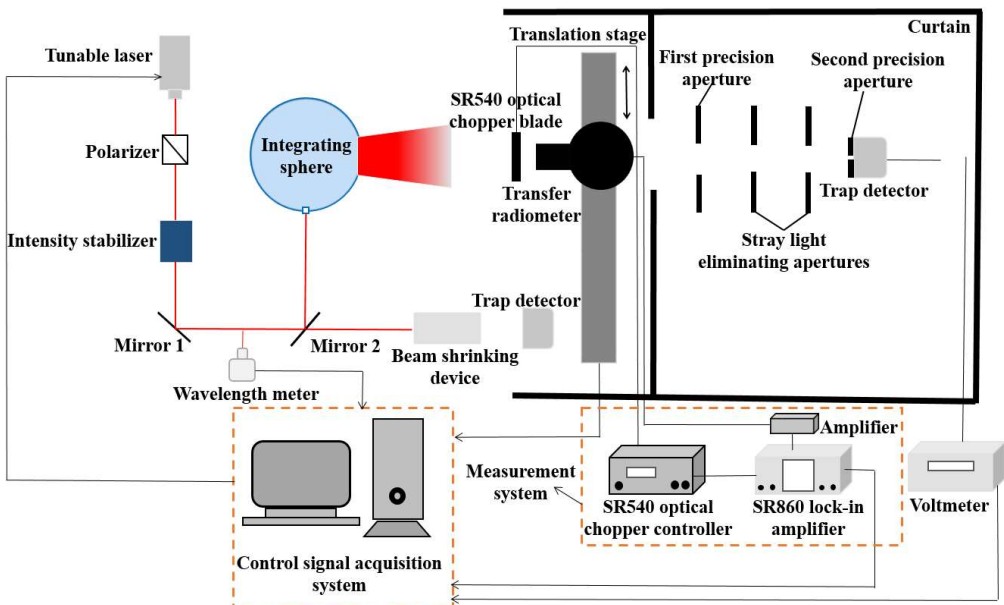

**Figure 1.** Experimental system for the TR power responsivity calibration and radiance measurement comparison.

**Table 1.** The TR filter-free channel power responsivities and radiance responsivities.

| Wavelength (nm) | Laser Power (mW) | Power Responsivity (V/W) | Radiance Responsivity ($10^{-8}$ V/(W/m$^2$/sr)) |
|---|---|---|---|
| 780.1 | 0.8157 | 0.01437 | 1.5767 |
| 851.9 | 0.8985 | 0.01601 | 1.7566 |

### 3.2. Narrowband, Detector-Based, Radiance Measurement Comparison Experiment

The radiance meter of the NIM consists of two precision apertures, two stray light-eliminating apertures, and a trap detector. The NIM radiometer radiance responsivity was derived according to the trap detector power responsivity and the two precision apertures' system geometry relationship. The radiance value measured by the NIM radiance radiometer was derived according to the output value of the voltmeter and radiance responsivity. Figure 1 shows the experimental setup for the radiance measurement comparison experiment between the TR and NIM radiance meter. Adjustment of the laser wavelengths to 780.0 nm and 851.9 nm was based on the wavelength meter feedback. The stability and absolute accuracy of the wavelength meter were $\pm 0.1$ ppm and $\pm 0.3$ ppm. The laser beam emitted from the tunable laser was incident on the inner baffle of the integrating sphere after passing through the intensity stabilizer and being reflected by two plane mirrors.

The rotating baffle suppressed the laser speckle effect. The TR was located between the integrating sphere and the NIM radiance meter, ensuring that the TR, the NIM radiance meter, and the integrating sphere exit optical axes coincided. The front aperture of the TR was placed 275 mm from the integrating sphere exit port. The TR measured the radiance of the integrating sphere exit port using the filter-free channel. The measurement system consisting of an SR860 lock-in amplifier, an SR540 optical chopper, and an amplifier measured the output signal of the TR. The TR filter-free channel radiance responsivity values were known as 780.0 nm and 851.9 nm. When the output voltage of the measurement system was obtained, the radiance value measured by the TR could be derived. Then, the NIM radiance meter measured the radiance of the integrating sphere exit port. The voltmeter measured the output signal of the NIM radiance meter. According to the radiance responsivity of the NIM radiance meter and the output value of the voltmeter, the radiance measured by the NIM radiance meter was obtained. Table 2 shows the radiance measurements of the TR filter-free channel and the NIM radiance meter, as well as the measurement differences between the two instruments.

**Table 2.** The TR and NIM radiance meter radiance measurement results and the measurement differences at 780.0 nm and 851.9 nm.

| Wavelength (nm) | TR Radiance Measurement Result (W/m$^2$/sr) | NIM Radiance Meter Radiance Measurement Result (W/m$^2$/sr) | Relative Difference (%) |
|---|---|---|---|
| 780.0 | 0.2415 | 0.2414 | 0.04 |
| 851.9 | 0.2319 | 0.2320 | −0.04 |

### 3.3. Uncertainty Analysis

A prior study [20] documented the uncertainty analysis of the radiance measurement comparison experiment for the TR filter-free channel. The relative measurement uncertainty of the NIM radiance meter is 0.3% (k = 1). Compared with the previous TR filter-free channel radiance measurement comparison experiment [20], this study changed the measurement system; the relative measurement uncertainty components of this TR filter-free channel radiance measurement comparison experiment were the same as the aforementioned experiment except for the measurement system. Subsequently, we mainly present the measurement uncertainty introduced by the new measurement system. The coverage factors of all uncertainty components were k = 1. The measurement system consisting of an SR860 lock-in amplifier, an SR540 optical chopper, and an amplifier measured the output signal of the TR in the power responsivity calibration and the radiance measurement comparison experiments. The output voltage of the measurement system of the TR was traceable to the trap detector. Although the signal magnitude differed between the two experiments of the TR, the measurement system of the TR remained the same, i.e., the magnification of the amplifier primarily determined the relative measurement uncertainty of the TR measurement system. In the two experiments of the TR power responsivity calibration and radiance comparison, the signals received by the TR Si detector differed by four orders of magnitude, and the relative measurement uncertainty introduced by the amplifier was 0.1%. Thus, the relative measurement uncertainty introduced by the measurement system of the TR was 0.1%. Referring to the previous results [20], the relative measurement uncertainty introduced by the other uncertainty components of the TR filter-free channel was 0.21%. The relative measurement uncertainty introduced by the TR filter-free channel was 0.24% in the radiance measurement comparison experiment.

The normalized error was used to evaluate whether the measurement results of the TR were consistent with those of the NIM radiance meter. The normalized deviations can be expressed as:

$$|E| = \left| \frac{L_{TR} - L_{NIM}}{k \cdot \sqrt{(L_{TR} \cdot \sigma_{TR})^2 + (L_{NIM} \cdot \sigma_{MIN})^2}} \right| \tag{1}$$

where $L_{TR}$ is the radiance measured by the TR, $L_{NIM}$ is the radiance measured by the NIM radiance meter, $\sigma_{TR}$ is the relative measurement uncertainty of the TR, and $\sigma_{NIM}$ is the relative measurement uncertainty of the NIM radiance meter, with a unity coverage factor k = 1.

Incorporating the relevant parameters, the results of the normalized errors at 780.0 nm and 851.9 nm can be derived as: $|E_{780.0}| = 0.11 < 1$ and $|E_{851.9}| = 0.11 < 1$. The results show that the radiance measurement results of the TR were consistent with those of the NIM radiance meter. It can be assumed that the measurement uncertainty of the TR filter-free channel is about 0.24%.

*3.4. Spectral Radiance Responsivity Curve Measurements of the Transfer Radiometer with 780.4 nm and 851.8 nm Filter Channels*

The experimental overview for the spectral radiance responsivity measurements is shown schematically in Figure 1. The tunable laser was adjusted to a specific wavelength by wavelength meter feedback. The wavelength adjustment range of the TR 780.4 nm filter channel was between 761.2 nm and 800.1 nm, the wavelength adjustment range of the TR 851.8 nm filter channel was between 830.2 nm and 871.5 nm, and the wavelength adjustment interval of the two filter channels was approximately 2 nm. Firstly, the output laser beam of the tunable laser, transited by the intensity stabilizer, was reflected by plane mirror 1 and was then received by the trap detector. The laser power value was obtained according to the output voltage of the trap detector and its power responsivity. When the trap detector completed its measurement, it was removed. The TR measured the laser beam using the filter-free channel. The measurement system measured the output voltage of the TR, and the laser power value was recorded; thus, the TR filter-free channel power responsivity was derived according to the above two values. The TR power and radiance responsivity transformation coefficient was recorded, and the TR radiance responsivity could be obtained according to the transformation coefficient and the power responsivity. Following the rotation of plane mirror 2, the output light from the tunable laser entered the integrating sphere after passing through the intensity stabilizer. The TR was set successively to the filter-free channel, the 780.4 nm filter channel, or the 851.8 nm filter channel to measure the quasi-Lambert source emanating from the integrating sphere exit port. The transmittance of the filter with a center wavelength of 780.4 nm or 851.8 nm at a certain specific wavelength was obtained according to the output voltage of the filter-free channel, the 780.4 nm filter channel, or the 851.8 nm filter channel of the TR. The radiance responsivity of the 780.4 nm filter channel or 851.8 nm filter channel at a particular wavelength was derived from the radiance responsivity of the filter-free channel and the corresponding filter transmittance. The spectral radiance responsivity curves of the 780.4 nm filter channel and 851.8 nm filter channel of the TR were obtained by changing the tunable laser wavelength successively and repeating the above steps, according to the above wavelength adjustment range and wavelength adjustment interval of the two filter channels. Figures 2 and 3 show the measurements of 780.4 nm and 851.8 nm filter channel spectral radiance responsivities, respectively.

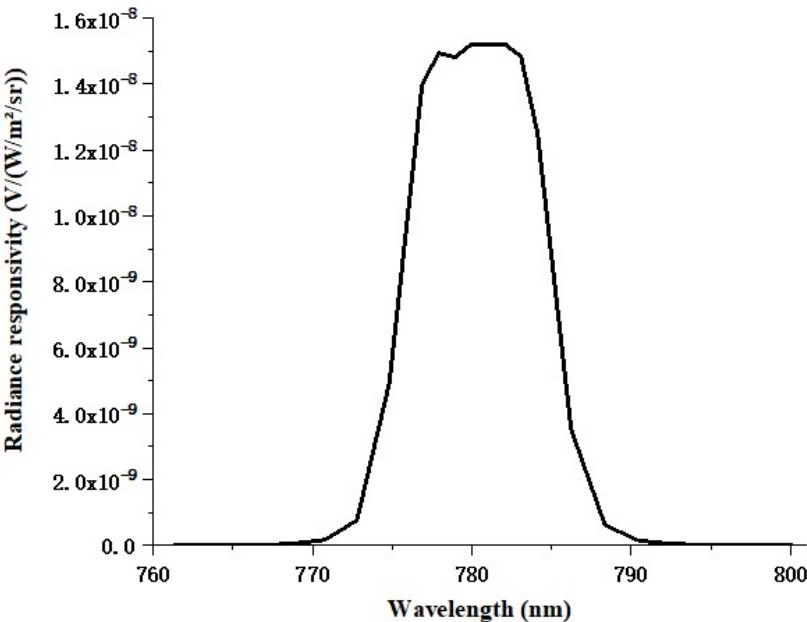

**Figure 2.** The spectral radiance responsivity curve of the TR 780.4 nm filter channel.

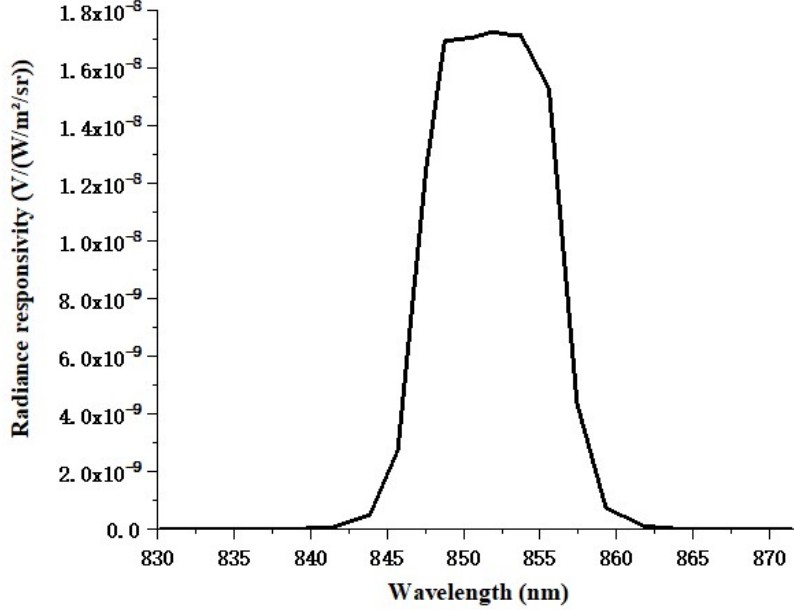

**Figure 3.** The spectral radiance responsivity curve of the TR 851.8 nm filter channel.

## 4. Broadband, Source-Based, Radiance Measurement Comparison Experiment

We selected the TR 780.4 nm and 851.8 nm filter channels to complete the broadband radiance comparison experiment between the TR and the integrating sphere source calibrated by the NIM. The experiment goal was to validate the precision of the TR by measuring the radiance of a broadband light source, such as a halogen tungsten lamp.

The experimental setup is shown in Figure 4. Halogen tungsten lamps were situated inside the integrating sphere. The diameter of the integrating sphere source was 1 m, and the exit port diameter of the integrating sphere source was 260 mm. The TR front aperture was placed 500 mm from the integrating sphere source exit port. The integrating sphere source was turned on, and the TR successively measured the quasi-Lambert source emanating from the integrating sphere source exit port using the 780.4 nm filter channel and 851.8 nm filter channel. The measurement system was composed of the SR860 lock-in amplifier, the SR540 optical chopper, and the amplifier which measured the TR output

signal. The SR860 lock-in amplifier and SR540 optical chopper settings were the same as those detailed in Section 3. Assuming that the spectral radiance responsivity of the TR filter channel at a certain wavelength was $R(\lambda_i)$, and the output value of the TR was $V$, the radiance of the integrating sphere source measured by the TR can be expressed as:

$$L_{\lambda_m} = \frac{V}{\int_{\lambda_1}^{\lambda_n} R(\lambda_i)d\lambda} \tag{2}$$

where $\lambda_m$ is the central wavelength of the spectral radiance responsivity curve of the TR filter channel, and $\lambda_1$ and $\lambda_n$ are the passband wavelength bounds of the corresponding wavelength range of the spectral radiance responsivity curve of the TR filter channel.

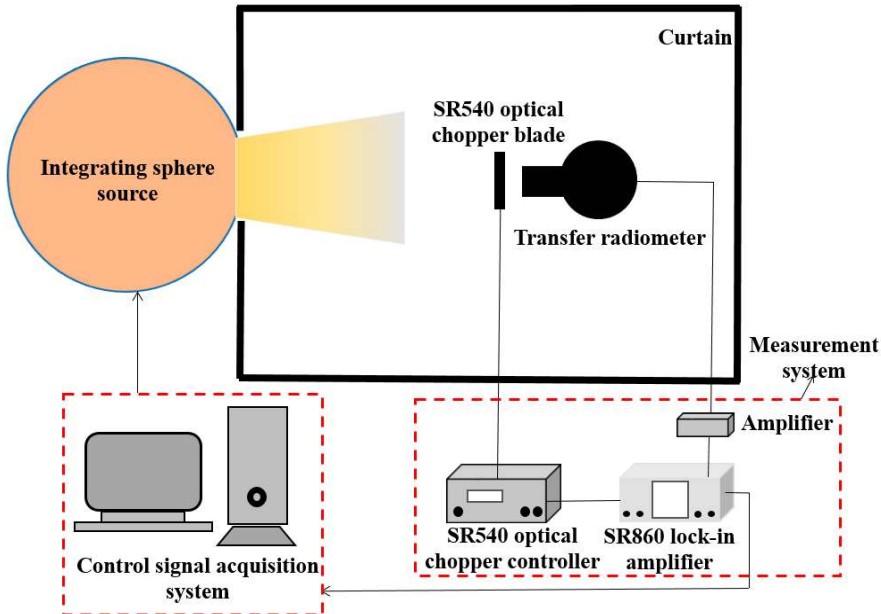

**Figure 4.** Experimental system of the TR broadband, source-based, radiance measurement comparison.

In Section 3, the spectral radiance responsivity curves of the 780.4 nm and 851.8 nm filter channels of the TR are presented. According to the output voltages of the two TR filter channels, the measuring radiance values of the two TR filter channels at 780.4 nm and 851.8 nm could be derived from Equation (2). The integrating sphere source radiance values at 780.4 nm and 851.8 nm could be derived by interpolation. Table 3 shows the deviations between the two TR filter channels measuring radiance values and the integrating sphere source radiance values at 780.4 nm and 851.8 nm.

**Table 3.** Relative differences between the results of the two TR filter channels measuring radiance and the integrating sphere source radiance values at 780.4 nm and 851.8 nm.

| Wavelength (nm) | TR Radiance Measurement Result (W/m$^2$/sr) | Integrating Sphere Source Radiance (W/m$^2$/sr) | Relative Difference (%) |
|---|---|---|---|
| 780.4 | 0.0302 | 0.0303 | −0.33 |
| 851.8 | 0.0354 | 0.0356 | −0.56 |

The measurement uncertainty sources in TR broadband, source-based, radiance measurement comparison experiments were mainly from the TR and the integrating sphere source. The radiance of the integrating sphere source was calibrated by the NIM. The relative measurement uncertainty of the integrating sphere source was provided by the

NIM, and the relative measurement uncertainty was 1.95% (k = 1) in the wavelength range of 700 nm to 900 nm. On the other hand, the relative uncertainties of the TR 780.4 nm and 851.8 nm filter channel measurement were mainly affected by spectral radiance responsivities of the two TR filter channels, stray light, nonlinearity of the detector, measurement repeatability, uncertainty of the measurement system, light leakage from the 780.4 nm and 851.8 nm filters, and temperature.

(1) The Relative Measurement Uncertainty of the TR Filter Channel Spectral Radiance Responsivity

Similar to the TR filter-free channel, the relative measurement uncertainty of the spectral radiance responsivity of the TR filter channel was mainly affected by power measurement repeatability $u_1(\lambda_i)$, trap detector power responsivity calibration uncertainty $u_2(\lambda_i)$, front aperture diameter $u_3(\lambda_i)$, rear aperture diameter $u_4(\lambda_i)$, the distance between the front and rear apertures $u_5(\lambda_i)$, the eccentricity $u_6(\lambda_i)$ and parallelism $u_7(\lambda_i)$ of the front and rear apertures, radiance measurement repeatability $u_8(\lambda_i)$, the nonlinearity of the detector $u_9(\lambda_i)$, stray light $u_{10}(\lambda_i)$, and the uncertainty of the measurement system $u_{11}(\lambda_i)$.

The front and rear apertures of the TR were composed of stainless steel, which is little affected by temperature. Except for the front and rear apertures, all other parts of the radiance-measuring tube were composed of aluminum. The distance between the front and rear apertures was affected by temperature. The diameters of the front and rear apertures were measured using a universal tool microscope; the measurement uncertainties of diameters of the front and rear apertures were 0.04% (k = 1) and 0.08% (k = 1), respectively. The distance between the front and rear apertures and the tilt and eccentricity between the front and rear apertures were measured using a coordinate measuring machine. Considering the effect of a temperature change of 5 °C on aluminum length, the measurement uncertainty of the distance between two apertures was 0.04% (k = 1). The measurement uncertainties introduced by tilt and eccentricity were 0.0001% (k = 1) and 0.0002% (k = 1), respectively. The above uncertainties were determined by the TR itself, independent of the wavelength. Power measurement repeatability and radiance measurement repeatability were mainly affected by the TR and the source. The trap detector measurement uncertainty was provided by the NIM. The signals of 780.4 nm and 851.8 nm filter channels of the TR were collected by a Si detector, and nonlinearity of the detector was affected by the measured signals. The measured signals of the two TR filter channels were of the same magnitude; therefore, the uncertainty introduced by nonlinearity of the detector was similar. The TR suppressed stray light using a radiance-measuring tube coated with black paint. In the wavelength range of approximately 700 nm to 900 nm, the absorption rate of the black paint is almost the same. The experimental source was a monochromatic laser. It was considered that the influence of the stray light on the two filter channels was similar. The voltage values measured by the TR measurement system were traced by the trap detector of the NIM. The measurement uncertainty introduced by the measurement system was mainly affected by the amplifier. Affected by the transmittance of the filters, the signal intensities received by the TR filter channels were different at different wavelengths; the above uncertainty components were related to the wavelength. The ambient temperature of the spectral radiance responsivity curve measurement experiment of the TR varied by approximately 0.2 °C; therefore, the influence of temperature could be ignored. It was assumed that the ratio of spectral radiance responsivity $R(\lambda_i)$ at a wavelength and total responsivity of the TR filter channel was $p_i$, and the specific values of each uncertainty component were incorporated at different wavelengths. The relative measurement uncertainties of spectral radiance responsivity of the TR 780.4 nm and 851.8 nm filter channels can be derived as follows:

$$u_{L_{780.4}} = \sqrt{\sum_{i=761.2}^{800.1} \sum_{m=1}^{11} p_i{}^2 \cdot u_m{}^2(\lambda_i)} = 0.68\%$$

$$u_{L_{851.8}} = \sqrt{\sum_{i=830.2}^{871.5} \sum_{m=1}^{11} p_i{}^2 \cdot u_m{}^2(\lambda_i)} = 0.71\%$$

(2) The Relative Measurement Uncertainties of Stray light, the Nonlinearity of the Detector, Radiance Measurement Repeatability, and the Measurement System

In order to determine the effect of stray light when the TR measured the integrating sphere source more accurately, we performed further experimental verification. A baffle was placed between the TR and the integrating sphere source; the baffle blocked the field of view (FOV) of the TR to ensure that only the light outside the FOV of the TR could be received. If there is no stray light, the output voltage value of the TR should be zero. However, the actual stray light proportion of the TR 780.4 nm filter channel was 0.07% (k = 1), and that of the TR 851.8 nm filter channel was 0.09% (k = 1).

In the broadband, source-based, radiance comparison experiment and spectral radiance responsivity curve measurement experiments of the TR 780.4 nm and 851.8 nm filter channels, the signals received by the TR Si detector differed by five orders of magnitude. The signals of the 780.4 nm and 851.8 nm filter channels of the TR were all collected by a Si detector, and the measurement uncertainties due to Si detector nonlinearity were 0.2% (k = 1). The TR measured the integrating sphere source for about 1 min. The radiance measurement repeatability values of the TR 780.4 nm and 851.8 nm filter channels are 0.12% (k = 1) and 0.08% (k = 1), respectively. The signals' magnitude is different between the TR broadband, source-based radiance comparison experiment and spectral radiance responsivity curves measurement experiments of the TR 780.4 nm and 851.8 nm filter channels. However, the TR measurement system remained the same; thus, the relative uncertainty of the measurement system was 0.2% (k = 1).

(3) Temperature Effects

The ambient temperature changed approximately 5 °C in the broadband, source-based, radiance comparison experiment and spectral radiance responsivity curve measurement experiments of the TR 780.4 nm and 851.8 nm filter channels. The Si detector and filters are affected by temperature. The impact of temperature on a Si detector in the wavelength range of 700 nm to 900 nm is 0.05%. The 780.4 nm and 851.8 nm filters were customized by the same manufacturer, the film material and coating process are the same, and the influence of temperature on the two filters was similar. The impacts of temperature on the TR 780.4 nm and 851.8 nm filters were 0.25% (k = 1).

(4) Light Leakage of the TR 780.4 nm and 851.8 nm filter channels

Light leakage of the TR filter channel was mainly affected by the filter. The filter transmittance curves were recorded. The ratio of the out-of-band transmittance of the filter to the in-band transmittance was used as the leakage ratio of the TR filter channel. For the TR 780.4 nm filter channel, the TR could receive light signals out of the wavelength range from 761.2 nm to 800.1 nm. The out-of-band rejection ratio of the 780.4 nm filter was of a $10^{-5}$ magnitude in the ranges of 300 nm to 761.2 nm and 800.1 nm to 1200 nm. The effect of light leakage out of the wavelength range of 761.2 nm to 800.1 nm of the TR 780.4 nm filter channel was 0.23% (k = 1). Similarly, for the TR 851.8 nm filter channel, the out-of-band rejection ratio of the 851.8 nm filter was of a $10^{-5}$ magnitude in the range of 300 nm to 830.2 nm and 871.5 nm to 1200 nm. The effect of light leakage out of the wavelength range from 830.2 nm to 871.5 nm of the TR 851.8 nm filter channel was 0.2% (k = 1).

(5) Relative Measurement Uncertainties of the Broad Radiance Responsivity

As shown in Equation (2), the integral sum of the radiance responsivity at each wavelength was used as the TR broadband radiance responsivity. The numerical integration method was used to calculate the integral sum of the radiance responsivity of the TR. The measurement uncertainty introduced by numerical integration was associated with the wavelength adjustment interval. The wavelength adjustment interval of the TR 780.4 nm and 851.8 nm filter channels was approximately 2 nm. The measurement uncertainties introduced by numerical integration of the TR 780.4 nm and 851.8 nm filter channels were

0.35% (k = 1) and 0.1% (k = 1), respectively. Table 4 shows the relative measurement uncertainty estimations of the two TR filter channels.

**Table 4.** Relative uncertainties estimation of the two TR filter channels.

| Uncertainty Origin | 780.4 nm Filter Channel (%) (k = 1) | 851.8 nm Filter Channel (%) (k = 1) |
| --- | --- | --- |
| TR filter channel spectral radiance responsivity | 0.68 | 0.71 |
| Radiance measurement repeatability | 0.12 | 0.08 |
| Stray light | 0.07 | 0.09 |
| Detector response linearity | 0.2 | 0.2 |
| Measurement system | 0.2 | 0.2 |
| Temperature | 0.25 | 0.25 |
| Light leakage of filter channels | 0.23 | 0.2 |
| Broad radiance responsivity | 0.35 | 0.1 |
| Combined uncertainty | 0.89 | 0.84 |

Normalized error was also used to evaluate whether the radiance measurement results of the TR were consistent with the integrating sphere source radiances at 780.4 nm and 851.8 nm. The normalized errors of the TR 780.4 nm and 851.8 nm filter channels can be derived from Equation (1), with the following results: $|E_{780.4}| = 0.15 < 1$, $|E_{851.8}| = 0.26 < 1$. The results show that the radiance measurement results of the TR 780.4 nm and 851.8 nm filter channels were consistent with the integrating sphere source radiances at 780.4 nm and 851.8 nm. It could be assumed that the measurement uncertainties of the TR 780.4 nm and 851.8 nm filter channels were 0.89% (k = 1) and 0.84% (k = 1), respectively.

## 5. Conclusions

The experimental results of the TR radiance comparison experiments show that the monochromatic source radiance measurement uncertainty of the TR is 0.24% (k = 1) at 780.0 nm and 851.9 nm, and the broadband source radiance measurement uncertinies of the TR 780.4 nm and 851.8 nm filter channels are 0.89% (k = 1) and 0.84% (k = 1), respectively.

A radiance comparison experiment was carried out between the TR and the NIM radiance meter. The TR and the NIM radiance meter alternately measured the same integrating sphere, which was illuminated by a laser beam. The relative measurement uncertainty of the TR is 0.24% (k = 1) at 780.0 nm and 851.9 nm. Additionally, at 780.0 nm and 851.9 nm, the radiance measurement results of the TR are consistent with those of the NIM radiance meter. The relative differences in radiance results between the TR and the NIM radiance meter are 0.04%.

Additionally, a broadband radiance comparison experiment between the TR and the integrating sphere source calibrated by the NIM was carried out. The TR measuring radiance of the integrating sphere source was obtained according to its radiance responsivity and output voltage. The relative measurement uncertainties of the TR 780.4 nm and 851.8 nm filter channels were 0.89% (k = 1) and 0.84% (k = 1), respectively. Additionally, the radiance measurement results of the TR 780.4 nm and 851.8 nm filter channels were consistent with the integrating sphere source radiances at 780.4 nm and 851.8 nm. The relative differences between the two TR filter channels measuring radiances of the integrating sphere source and the radiances of the integrating sphere source itself were better than 0.56%.

The EMIS calibrated by the TR through the BTC can measure radiance with an uncertainty of better than 2%. Although the target radiance-measuring uncertainty of better than 1% for the EMIS was not achieved, related studies of the TR are important for subsequent development of the project. Further research will be conducted on the TR to improve the uncertainty of radiance measurements so as to achieve an uncertainty of better than 1% in the radiance calibration of remote-sensing instruments such as the EMIS.

**Author Contributions:** Conceptualization, K.L. and X.Y.; methodology, K.L.; validation, K.L., S.L., N.X. and Z.L. (Zhiwei Liu); investigation, K.L. and Y.Z.; data curation, K.L., Z.L. and Z.L. (Zhiwei Liu); writing—original draft preparation, K.L.; writing—review and editing, Z.L. (Zhigang Li) and X.Y.; visualization, Y.W.; project administration, X.Y.; funding acquisition, X.Y. All authors have read and agreed to the published version of the manuscript.

**Funding:** This research was supported by the Strategic Priority Research Program of the Chinese Academy of Sciences under grant XDA28050102 and the National Key R&D Program of China under grant 2018YFB0504600, 2018YFB0504603.

**Institutional Review Board Statement:** Not applicable.

**Informed Consent Statement:** Not applicable.

**Data Availability Statement:** Not applicable.

**Conflicts of Interest:** The authors declare no conflict of interest.

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
