# Peer review of "Calibration and Validation of a Transfer Radiometer Applied to a Radiometric Benchmark Transfer Chain"

_photonics, doi:10.3390/photonics10020173_

Round 1

Reviewer 1 Report

The introduction requires improvement.  For example, why is it important to achieve the high level of accuracy?  This is clearly presented in the references (see ref. 10 in particular)  but not at all motivated in the paper, which would only take a few sentences.  The calibration relies upon work done at the National Institute of Metrology (NIM).  Why isn't there a reference to this?  For example, the SIRCUS uncertainties are traceable to POWR with the references provided in the SIRCUS publications enabling a full understanding of the uncertainty analysis.  For the error analysis, what is the impact of the interpolation going from the monochromatic measurement of the spectral radiance responsivity curve to the broad band measurement?  How are you certain that there are no leakages in the filters far from the nominal transmission wavelengths?  Were there any humidity changes in the laboratory as both bands overlap with water features and even the small overlap for these measurements may impact the transfer radiometer characterization.  This is expected to be small.  Each of the uncertainties (u1 through u11) could be better explained and u11 in particular should be addressed.  There is little connection between the paper and several of the references.  Some of the references have the authors being addressed by their first names rather than their last.

Reviewer 2 Report

Overall value of publication expands upon the earlier result form the development of a compact transfer radiometer for radiometric benchmark transfer chain (Lei, et al., Sensors, 2022, 22, 6795).  Application to the specific wavelength transfer uncertainties are documented in the calibration/validation. Overlap of measurement equations from power to radiance and are not duplicated in the current document but are referenced to the previous. Current submission provides the specific detail of the quantification of the system to meet spectral calibration requirements of climate benchmark radiometry. This compliments the previous efforts without duplication results and therefore provides for a stand-alone effort based on previous achievements.

Review of this submission brings up specific concerns, for this reviewer, that relate to areas of uncertainty in the full discussion related to the power responsivity and ties to radiance responsivity issues related to:

•uncertainties of the non-circular aperture errors and aperture deviation from circular for area uncertainties

•Diffraction considerations in the baffles and apertures (mostly common mode in direct comparisons, but changes due to configuration changes).

• Filter tilt and Fresnel losses (again may result in common mode in relative comparisons, but no discussion in text.

•Temperature changes for the length of separation of the aluminum for length (temperature stability requirements needed in discussions)

•Si and InGaAs radiation damage on-orbit (long-term stability maintainance for calibration ties over mission durations, not focus of this publication, but to be considered)

•Some care in the application of Lock-in detection across voltage measurement comparisons related to Vrms vs. Vpp issues in the phase sensitive detection.

Would recommend adding some text relevant to the uncertainty analysis with consideration to these appropriate effects (if appropriate) within the text sections related to the specific spectral calibration transfer (see specific comments in the following document line references).

Specific recommendations for corrections and clarifications (referenced here by paper line numbers):

*Editing and style specific comments should be viewed as recommendations*.

Line 19:  “exactitude” , better to use “absolute accuracy” ; also throughout the document there are inconsistencies in the spacing of the “nm” from the numeric values…please check the consistency of the spaces between the units and the numeric values (e.g. some references to 780.0 and 851.8 have “nm” with and without a space separating. Check throughout.

Lines 36-40: would recommend omitting these general statements and start the paper with the sentence of line 41 and add a few statements of context as related to the present approach and value of the present study over (or building on) those discussed in Ref 27.

Line 48: recommend changing wording to “…Which degrades the performance of remote sensors ultimately reducing the measurement accuracy.”

Line 51: recommend replace “are solar diffuse…” with “rely on solar diffusers and standard lamps.”

Line 57: “…including CLARREO and TRUTHS, respectively, which can…”

Lines 58-61: recommend omit first sentence “Chinese expert group…” and begin with “The Chinese Space-Based Radiometric Benchmark (CSRB)…Line 61 sentence that references “The Space Cryogenic Absolute Radiometer (SACR)…”  The acronym here is incorrect and not consistent with Ref 27, should be “SCAR”, please check and correct throughout the document.

Line 63: “…the reflected solar spectrum radiance, should read “the reflected solar spectral radiance”

Lines: 80-90:  Please add reference to Ref 27 and provide context for the support for the present measurements.

Line 85:  Here (and throughout the document), try to avoid using possessive reference to objects, i.e. avoid “transfer radiometer’s” and simply use “transfer radiometer”.  Also would recommend referencing the Transfer Radiometer” as “TR” in the remainder of the text.

Line 93:  “SACR” change to “SCAR”

Line 100:  add space after “0.06%.”

Line 104: recommend “The spectral radiance responsivity of the transfer radiometer filter channel is derived through the..”

Line 109: “…such as a halogen tungsten lamp.”

Line 119:  Confusing using wording like “…chopper stopped working” better to use “…was turned off”. Using “stopped working” seems to imply that it broke.

Line 126:  “SACR” …change to “SCAR”

Line 131: “…changed the integrating sphere”, recommend changing to “…replaced the integrating sphere”

Lines 133-143: This description would be better to reference a figure that shows an overview the heart of this description. The paragraph read like a descriptive “receipe” and without a graphic can be confusing.  If this references Figure 1 then it is recommended to move this description to the next section of the paper that references Figure 1.

Line 148: again terms like “did not work” implies broken or defective, recommend changing wording to “…the plane mirror 2 was removed from the optical path and…”

Line 149:  First mention of this “beam shrinking device”, is there  a reference?  Or recommend providing a brief discussion that would give the reader and understanding of the considerations of such device, for example, if lenses then what of scatter and aberrations, or mirrors and coating considerations for wavelengths used.

Lines 152-156: Can omit first sentence. Then “Adjusting the tunable wavelength..”  Also, there is no discussion of the type of laser or method of tuning for these lasers (also could not find discussion in Ref 27). Ti:Sapphire or diode? Pulsed or CW?, mode structure? Line widths?  This has consequences for the spectral responsivity curves in Figure 2 & 3 with respect to deconvolution of laser line width influence in the responsivity curves. I suspect that this is not a problem due to laser having narrow linewidth, but a brief discussion would help to clarify.

Line 165-167: recommend changing wording to “ The transfer radiometer filter free channel power responsivity was derived from the measured output voltage as a function of measured laser power.”

Line 175: “radiometer radiance”

Line 176: “detector power”

Line 180-181:” recommend wording “The adjustment of the laser wavelengths to 780.0 nm and 851.9 nm was based on the wavelength meter feedback”.  Also, at what stability?  Would suggest referencing the laser wavelength meter that discussed the stability parameters.

Line 185: change “and ensure” to “ensuring”

Line 187: delete “away” after 275mm, reads “…was placed 275 mm from the…”

Line 191: “channel’s” to “channel”

Lines 194-196: recommend delete the sentence “The transfer radiometer should…”

Table 2: What are the associated uncertainties of the measurement results here for each of the spectral radiance values, For example 0.2415 ± ?  That value should be used (in quadrature) to bound the relative difference values. Was this discussed elsewhere, if so then add to table.

Section 3.3, Line 210:  first sentence can delete the words “has been”,  should read “Reference [27] introduced (better to us “documented”) the uncertainty analysis…”

Line 212 (and throughout text): Avoid qualifying uncertainties with words like “about” or “approximately” when hard quantitative values (with k) are referenced as determined uncertainties.

Line 216-217:  Reference to “k” are not strictly confidence levels, but (as per NIST definitions) are uncertainty “Coverage factors” to meet the requirements of an expanded uncertainty; the use of confidence levels typically is in reference to normal distributions and statistical analysis and probability distributions that can be difficult to uniquely quantify for certain measurement uncertainties.

Lines 220-227: description here is a bit “wordy” is the details, recommend tightening up the texrt to reflect the specifics with fewer words.

Line 244: recommend “the unity coverage factor k=1”

Section 3.4

Line 252: First sentence should be more descriptive, recommend “ The experimental overview for the spectral radiance responsivity measurements is shown schematically in Figure 1.”

Line 261-262: Delete the sentence “The measurement system…” and add it to the description of the following sentence to add clarity.

Line 263 (and 265): avoid “known” and use “recorded” or “derived”

Line 267: recommend changing sentence to “ Following the rotation of plane mirror 2, the output light form the tunable laser entered the integrating sphere after passing through the intensity stabilizer.”

Line 280-281: recommend sentence read “Figures 2 and 3 show the measurement 780 nm and 850 nm filter channel spectral radiance responsivities, respectively.”

Figures 2 & 3 and Equation 2: Some details about the spectral sampling would be in order. Unclear from the plots as to the definition of the spectral structure seen. For example, a brief discussion on the laser tuning intervals required to achieve the level of uncertainty in the radiance responsivity definitions. A sentence or two would suffice if it is not a limiting factor to the reference to the integral in Equation 2 by use of the measured spectral radiance responsivities and the measured output voltages of the transfer radiometer response at a given wavelength.

Line 307: change “upper and lower limits” to “passband wavelength bounds”

Line 310: Change “Chapter 3” to “Section 3”

Lines 324-325: Where is the integrating sphere source uncertainty of 1.95% documented? Is this discussed in another reference (couldn’t find reference to this in Ref [27]. Need some supporting information either in this section where the uncertainties become propagated to derive the combined standard uncertainties. Why was this not included in the full uncertainty budget, some clarification would be goo in the text with reference to the final results of Table 4 later.

Line 333:  recommend changing “Take the transfer…” to “Using the example of the 780.4 nm filter channel, …”

Lines 333-353:  This discussion is a bit confusing with respect to the specific uncertainties for each channel (780 and 850 nm). It is a bit surpising that all unit uncertainties in the error budget are identical for both filter channels given the changes in the responsivity values and laser measurement parameters (e.g. scattered and stray light contribution differences?  The following section discussion has some brief discussion with respect to stray light and detector non-linearity, but does not provide any wavelength specific differences. Can the text provide further clarification based on the values listed in Table 4?, if so, then recommend some added discussion relevant to the uncertainty magnitudes (and lower bounds).  Reference is made to these effects being “superior to” some uncertainty limits, this is a qualitative assessment rather than a unique quantitative comparison.

Line 383: recommend changing “excellent inhibition ability” to “high out-of-band rejection ratio, (quantify 10^-4? 10^-5?).  This is not discussed in detail in the text, can a reference be made here to any supporting measurements of the filters specific rejection performance, table or figure?)

Table 4: Again, a bit surprised that error budget are identical for both filter channels given the magnitude differences in the measured filter spectral response functions of Figures 2 & 3, recommend providing some discussion of this relevant to the wavelength specific sensitivities (integral errors in Equation 2 due to spectral sampling?)

Lines 405-417: Not sure of the relevance to the overall transfer radiometer calibration discussed. Arguments here seem somewhat speculative and would serve better in a footnote rather than a discussion within the text of the calibration.

Section 5

Lines 421 & 422: avoid “about” and state specific uncertainties as “0.24% (k=1)” and “0.83% (k=1)”, respectively.  Same with “approximately” in Lines 426 & 429. Recommend omit these descriptions.

Line 422: Omit “Firstly” and just state “The radiance comparison…”

Line 431: recommend changing “Secondly” to “Additionally”

Line 444: provide context if the accuracy of the radiance calibrations derived here actually meets the requirements for EMIS and other remote sensors to improve their retrievals. The use of “high-accuracy” here is a relative assessment and must be weighed against the complimentary measurement requirements of the full remote sensing suite.

References

Reference 16 (Line 477-478): Missing publication date (add Sept 2007)

Round 2

Reviewer 1 Report

Citation 9 must be rewritten. 
